# End-to-End Continuous/Discontinuous Feature Fusion Method with Attention for Rolling Bearing Fault Diagnosis

**DOI:** 10.3390/s22176489

**Published:** 2022-08-29

**Authors:** Jianbo Zheng, Jian Liao, Zongbin Chen

**Affiliations:** 1Institute of Vibration and Noise, Naval University of Engineering, Wuhan 430033, China; 2Naval Key Laboratory of Ship Vibration and Noise, Naval University of Engineering, Wuhan 430033, China

**Keywords:** fault diagnosis, rolling bearing, deep learning, LSTM, CNN, attention

## Abstract

Mechanical equipment failure may cause massive economic and even life loss. Therefore, the diagnosis of the failures of machine parts in time is crucial. The rolling bearings are one of the most valuable parts, which have attracted the focus of fault diagnosis. Many successful rolling bearing fault diagnoses have been made based on machine learning and deep learning. However, most diagnosis methods still rely on complex signal processing and artificial features, bringing many costs to the deployment and migration of diagnostic models. This paper proposes an end-to-end continuous/discontinuous feature fusion method for rolling bearing fault diagnosis (C/D-FUSA). This method comprises long short-term memory (LSTM), convolutional neural networks (CNN) and attention mechanism, which automatically extracts the continuous and discontinuous features from vibration signals for fault diagnosis. We also propose a contextual-dependent attention module for the LSTM layers. We compare the method with the other simpler deep learning methods and state-of-the-art methods in rolling bearing fault data sets with different sample rates. The results show that our method is more accurate than the other methods with real-time inference. It is also easy to be deployed and trained in a new environment.

## 1. Introduction

The fault diagnosis of mechanical equipment is vital in modern industry. Once the failure of mechanical equipment occurs, it will cause huge damage to the economy and property and even bring casualties. Therefore, finding a better fault diagnosis method is necessary to ensure the normal operation of the machine [1]. Rolling bearing is the most commonly used part in mechanical equipment, known as the joint of the machinery. It has the advantages of high efficiency, small friction resistance, convenient assembly and easy lubrication, so it is widely used in rotating machinery. As one of the core components of rotating machinery, such as gearbox and turbine machinery, the health of rolling bearings significantly influences the machine’s stability and life [2]. In the process of working, rolling bearings may be damaged by the outer raceway, inner raceway and rolling body due to lubricant pollution, overload and other reasons. Therefore, an effective fault diagnosis method is crucial to the stability of rolling bearings [3].

Most fault diagnosis methods used for rolling bearings are based on vibration signals. By detecting and analyzing the vibration data of rolling bearings, this method can diagnose various faults in real time [4]. Model-based [5] and data-driven [6] methods can diagnose rolling bearing faults based on vibration data. Data-driven methods generally use machine learning to learn bearing vibration data and identify different types of fault modes. This method can effectively and quickly process mechanical signals, requires less prior expertise, and can provide an accurate diagnosis [7]. It has become a common method for rolling bearing fault diagnosis. K-nearest neighbor (KNN), support vector machine (SVM), self-organizing mapping (SOM) networks and other machine learning algorithms have been successfully applied to the intelligent diagnosis of rolling bearing faults [8,9,10]. Fault diagnosis methods based on traditional machine learning need to extract the artificial features to enhance the diagnosis performance, such as time-domain statistical features (root mean square, kurtosis, skewness and spectral kurtosis) [11,12,13,14,15], fast Fourier transform (FFT) spectrum, power spectrum, empirical mode decomposition (EMD) features, variational mode decomposition (VMD) features and other features [16,17,18]. For example, Xiao et al. proposed a fault diagnosis method based on the kurtosis criterion VMD and an SOM neural network. The method used the VMD algorithm to decompose the gear vibration signal, selected the intrinsic mode functions (IMFs) most relevant to the original vibration signals according to the average instantaneous frequency, and extracted the kurtosis of IMFs as the feature. This feature has been proven to diagnose gear faults accurately [10]. Song et al. proposed a feature-extracting method for rolling bearing vibration fault signals combining statistical filtering (SF), wavelet packet transform (WPT) and the moving peak value preservation (M-PH) method, which could identify fault types based on bearing diagnostic features in the frequency domain. This decision tree trained on the features could accurately diagnose the fault types of rolling bearings [18]. However, most rolling bearing fault diagnosis algorithms based on traditional machine learning rely on complex and well-designed signal feature extraction methods to accurately diagnose. It is necessary to adapt the feature extraction algorithm for specific situations or bearing types to achieve an accurate diagnosis. Therefore, this diagnosis method is susceptible to specific rolling bearings, experimental environment and feature adaptability, and it is difficult to achieve a general rolling bearing fault diagnosis system [19,20].

In recent years, deep learning [21,22,23,24,25] has been successfully applied in computer vision [26], natural language processing [27], speech recognition [28], mechanical control [29] and many other fields. This kind of algorithm integrates feature engineering and pattern recognition through the combination of multilayer artificial neural network layers. It can extract suitable features adaptively without complex artificial features for most data training. Therefore, various model structures of deep learning algorithms have been applied to rolling bearing fault diagnosis and achieved good performance, including deep belief networks (DBN), fully connected networks, long short-term memory (LSTM) models [30], convolutional neural networks (CNNs), transformers and others [7,31,32,33,34,35,36]. These studies proposed deep learning models with different structures or multimodal features to diagnose bearing faults. For example, Shao et al. proposed an optimized DBN for rolling bearing fault diagnosis. They used the stochastic gradient descent algorithm based on the energy functions to fine-tune all of the connection weights of the constrained Boltzmann machine (RBM) after pretraining effectively. The method improved the classification accuracy of DBN. Furthermore, an optimal DBN was designed using the particle swarm optimization algorithm to determine the best structure of the trained DBN. The method was applied to the simulation signal and experimental signal analysis of a rolling bearing and achieved higher precision and robustness than other intelligent methods [31]. Chen et al. proposed a model composed of a multiscale CNN and an LSTM (MCNN-LSTM) for rolling bearing fault diagnosis. This model consisted of a feature extractor and a classifier, allowing raw data to be input directly into the model without preprocessing. The feature extractor contained two CNNs with different kernel sizes, which were used to automatically extract the feature representation of the rolling bearing fault vibration signals. The extracted features were then input into a stacked LSTM network for bearing fault assessment [36]. The proposed method achieved 98.46% accuracy, which exceeded some state-of-the-art algorithms. Ding et al. proposed a new time-frequency transformer (TFT) model. This model adopted synchrosqueezed wavelet transform (SWT) [37] to obtain time-frequency representation. They designed an encoder composed of transformer blocks to construct the hidden time-frequency representation for rolling bearing faults. The method was proven to be superior in comparison with other methods [7]. Although the above algorithms and methods have achieved satisfactory results, there are some limitations.

Most current studies still need various signal processing methods to extract features. Therefore, the effectiveness of fault diagnosis heavily depends on the quality of manually extracted features. A suitable intelligent diagnosis algorithm is needed for adaptive feature extraction and selection.Some deep learning algorithms still need to cooperate with many complex signal processing methods to adapt to rolling bearing fault detection. These methods have a lot of manual parameters to adjust and are difficult to deploy and train.Some deep learning models improve the diagnosis effect by combining overly complex structures, but this usually increases the cost of calculation and the risk of overfitting.

In order to improve the above limitations, this paper proposes an end-to-end continuous/discontinuous feature fusion method with CNN, LSTM and attention (C/D-FUSA) for rolling bearing fault diagnosis, which does not require complex data processing and feature engineering. The proposed method mainly combines multilayer CNN and a multilayer LSTM network for adaptive feature extraction and the fusion of different features. A multilayer CNN network extracts the discontinuous large-scale features of rolling bearing vibration signals with several one-dimensional convolutional layers and one-dimensional pooling layers. Multilayer LSTM networks extract continuous features with short- and long-term memory by composing multiple bidirectional LSTM layers. Rolling bearing faults can be diagnosed by the fusion of continuous and discontinuous features. This work has the following contributions:An end-to-end deep learning model is proposed for rolling bearing fault diagnosis. Without manual design features or complex data processing, this model can accurately extract and screen continuous/discontinuous signal features to diagnose rolling bearing faults.Compared with the simple deep learning model, the proposed model has a higher classification accuracy (99.87%), and the inference time does not significantly increase.This method could be easily deployed and migrated to a new environment or a new type of rolling bearing because of the absence of complex data processing and manual feature engineering.The proposed model can achieve accurate multitype fault diagnosis, and the experiment proved that it could accurately diagnose 10 types of working states of rolling bearings.

The rest of this article is arranged as follows. Section 2 introduces the background and the details of the C/D-FUSA approach proposed by this work. The experimental setup and results are described in Section 3. The experimental results are discussed in Section 4. Finally, we give the conclusion in Section 5.

## 2. Materials and Methods

### 2.1. Types of Rolling Bearing Faults

A typical rolling bearing structure is shown in Figure 1, including an outer race mounted on a bearing house, an inner race mounted on a rotating shaft, a rolling element, and a supporting cage [38]. Rolling bearings are the most easily damaged parts in rotating machinery. Any damage will lead to the rapid degradation of the bearing due to wear and tear of the inner ball. There are many reasons for bearing faults, including improper size selection, leakage, excessive load and others. Once a bearing has a fault, it will produce periodic abnormal vibrations, whose amplitude is determined by the type of fault. The faults could happen in the bearings at the drive end or fan end. In most situations, the fault happens at the inner raceway, rolling element (ball) and outer raceway of a bearing. The faults usually range from 0.007 inches to 0.040 inches in diameter [39]. The types of faults can be diagnosed by the frequency features calculated by fault size, shaft speed, the load, and fault location, such as the inner circle rotation frequency, relative rotation frequency of the inner and outer circles, cage rotation frequency and the revolution frequency of the rolling body [40]. Some of the vibration frequencies are listed as follows:(1)$$f_{i}=\frac{n}{60}$$
(2)$$f_{r}=f_{i}-f_{0}$$
(3)$$f_{c}=0.381-0.4\times f_{r}$$
(4)$$f_{b}=\frac{1}{2}\left(1-\frac{d}{D}cos\alpha \right)f_{r}$$
(5)$$f_{be}=\frac{1}{2}N_{b}\left(1-\frac{d}{D}cos\alpha \right)f_{r}$$
(6)$$f_{bi}=\frac{1}{2}N_{b}\left(1+\frac{d}{D}cos\alpha \right)f_{r}$$

The n is the speed of bearing inner circle; $f_{i}$ is the inner circle rotation frequency; $f_{0}$ is the outer circle rotation frequency; $f_{r}$ is the relative rotation frequency of the inner and outer circles; $f_{c}$ is the cage rotation frequency; $f_{b}$ is the revolution frequency of the rolling body; and $f_{be}$ and $f_{bi}$ are the outer race and inner race fault frequencies, respectively. The frequency domain features obtained by FFT can also help to diagnose faults. This was used in the cepstral analysis, which takes the logarithm of the Fourier transform of the original waveform and then takes the inverse Fourier transform of the logarithm [41]. The inverse Fourier transform results could be different for the normal and fault states, as shown in Figure 2.

In traditional methods, these features are transformed into specific numerical forms, such as kurtosis, mean and variance, and then fed into machine learning algorithms for classification. Although the frequency features summarized from experiences can be used to infer the specific fault types or locations of the rolling bearings in some cases, the fault diagnosis method based on manually designed features is difficult for widespread use. The end-to-end diagnosis model based on deep learning proposed in this work does not depend on any experience, data statistics or processing, and can directly infer specific fault types from raw data.

### 2.2. Framework of the Proposed C/D-FUSA

This paper proposes an end-to-end continuous/discontinuous feature fusion method with CNN, LSTM and attention (C/D-FUSA) for rolling bearing fault diagnosis. The framework of the proposed C/D-FUSA is shown in Figure 3. This method contains three subnetworks. (1) The subnet for the continuous features is designed to extract sequential associated continuous features using LSTM layers with a context-dependent attention mechanism. (2) The subnet for the discontinuous features is designed to extract different level discontinuous features using convolutional layers with an attention mechanism. (3) The continuous and discontinuous features are concatenated and used to calculate the probability of different fault or normal states of rolling bearings in the subnet for classification.

### 2.3. Subnet for Continuous Features

The subnet for the continuous features is composed of LSTM layers with an attention mechanism. We proposed a context-dependent attention module for weighting the output of the features by the LSTM layer according to previous outputs.

#### 2.3.1. Long Short-Term Memory

LSTM is a network with a long-term memory function that is improved based on the recurrent neural network (RNN) [42], as shown in Figure 4. Due to its characteristics, LSTM is widely used in fitting time series, which is very important for natural language processing, speech recognition, handwriting recognition and other applications. An LSTM cell comprises a forget gate, an input gate and an output gate. The forget gate controls information in the previous time steps remaining in the current cell state. The input gate controls the new information input into the current cell state. Finally, the output gate, based on the cell states, controls the output of the LSTM cell. These three gates are described in Equations (7)–(9):(7)$$f_{t}=\sigma \left(W_{f}^{T}\cdot \left[h_{t-1},x_{t}\right]+b_{f}\right)$$
(8)$$i_{t}=\sigma \left(W_{i}^{T}\cdot \left[h_{t-1},x_{t}\right]+b_{i}\right)$$
(9)$$O_{t}=\sigma \left(W_{o}^{T}\cdot \left[h_{t-1},x_{t}\right]+b_{o}\right)$$

In the above three equations, $f_{t}$ is the output of the forget gate at time step t; $i_{t}$ is the output of the input gate; $O_{t}$ is the output of the output gate; $W_{f}$, $W_{i}$, and $W_{o}$ are trainable weight matrices; $b_{f}$, $b_{i}$, and $b_{o}$ are the trainable bias in these gates; t−1 represents the previous time step; $h_{t-1}$ is the hidden state at the previous time step; $x_{t}$ is the input features at the time step t; and σ is the sigmoid function. After information filtering through the forget gate and input gate, the LSTM cell state is adjusted by Equations (10) and (11):(10)$$C_{t}=\tanh\left(W_{C}^{T}\cdot \left[h_{t-1},x_{t}\right]+b_{C}\right)$$
(11)$$c_{t}=f_{t}\times C_{t-1}+i_{t}\times C_{t}$$
where $C_{t}$ is the information for updating the cell state; $W_{C}$ is the trainable weight matrix; $b_{C}$ is the trainable bias; and $c_{t}$ is the updated cell state controlled by the forget gate and the input gate. Finally, the hidden state at the current time step is defined by:(12)$$h_{t}=O_{t}\times \tanh\left(c_{t}\right)$$

#### 2.3.2. Context-Dependent Attention

Since the output of the last time step of the LSTM network in C/D-FUSA was used for the fault diagnosis, the attention module based on time series was unavailable [43]. Therefore, this work proposes a context-dependent attention module for LSTM. The structure of the context-dependent attention module is shown in Figure 5. This module calculates the attention weights of the output features by integrating the outputs of L time points before the last time point. First of all, the average output features of the contextual outputs are calculated by:(13)om=∑t=T−LTOtL, om∈ℝn×1

In the equation, om is the average output features of the contextual outputs from the time step T−L to T; T is the number of time steps; L is the length of the contextual window; and n is the number of output features in the LSTM layer. Then, the average output features are converted into an attention weight vector by Equations (14) and (15):(14)al=g(Wl2×(Wl1×om))
where al is the attention weight vector; Wl1 is a trainable weight matrix, Wl1∈ℝk×n; Wl2 is a trainable weight matrix, Wl2∈ℝn×k; and g is the softmax function. The softmax function converts the result of the products to the range between 0 and 1. Finally, the output of LSTM is weighted by the attention weight vector by:(15)ol,T=aloT
where ol,T is the result vector of an LSTM layer with context-dependent attention; and oT is the output of the LSTM layer at the time step T. The T used in the proposed method was equal to the length of the input vibration signals. The number of output features n was 128.

### 2.4. Subnet for Discontinuous Features

The subnet for the discontinuous features is composed of several convolutional modules with attention. As shown in Figure 6, the convolutional module comprises the sequentially connected 1D convolutional layer, batch normalization operation, ReLU activation function, and squeeze-and-excitation (SE) attention module [44] and 1D max-pooling layer.

The convolutional layer in the convolutional module is calculated with the same padding and the kernel size is 3. The kernel of the max-pooling layer is 2. Therefore, for an $N\times M_{i}$ input matrix, the convolutional module could obtain an $\frac{N}{2}\times M_{o}$ output matrix, where N is the length of the input matrix and $M_{i}$ is the number of input features. $M_{o}$ is the number of output features.

The SE attention module is one of the most popular attention modules in computer vision. In this module, the input matrix is averaged first, as described in Equation (16):(16)Pm,j=∑i=1Npi,jN, 1≤j≤M
where Pm,j is the average value for feature j; Pi,j is the feature value of the input matrix; and Pm is the average vector, Pm∈ℝM×1. Then, the Cm is squeezed by:(17)Cs=ReLU(Ws×Pm+bs)
where Cs is the squeezed vector, Cs∈ℝS×1; Ws is the trainable weight matrix, Ws∈ℝS×M; and bs is the trainable bias. After the squeeze, the vector is then excited by:(18)Ce=σ(We×Cs+be)
where Ce is the excited vector, Ce∈ℝM×1; We is the trainable weight matrix, We∈ℝM×S; be is the trainable bias; and σ is the sigmoid function. Finally, the original output P of the previous layer is weighted by Ce:(19)Pa=P·Ce
where Pa is the output of the attention module.

The C/D-FUSA uses three convolutional modules in the subnet for the discontinuous features. In the first convolutional module, the number of output features in each module is 32, 16 and 8, respectively. The squeeze coefficient S in each module is 8, 4 and 2, respectively.

### 2.5. Subnet for Classification

In the subnet for classification, the outputs of the discontinuous and continuous features by convolutional modules and the LSTM module with attention are concatenated. However, the output feature number of the convolutional modules (T) was much larger than that of the LSTM module 128. Too many discontinuous features could impair the diagnosis performance. Therefore, the output of the final convolutional module oc is converted by:(20)oc′=σ(Wc×oc+bc)
where oc′ is a vector with 16 features and Wc is a trainable matrix, Wc∈ℝ16×T. After the concatenation of oc′ and ol,T, two fully connected layers were used for classification. The first fully connected layer converted the 144 concatenated features into 64 features. The last fully connected layer converted the 64 features into NC values. The NC is the number of categories. Finally, a softmax function converted these NC values into NC probabilities, ranging from 0 to 1. Each of the values represented the probability of a fault state or normal state of the rolling bearing.

## 3. Experiments and Results

### 3.1. Experiment Setups

We tested the proposed C/D-FUSA method on the experimental data set from the rolling bearing data center of Case Western Reserve University (CWRU) [45]. The CWRU data set was collected using a controllable motor of 2 HP (power: 1.5 kW), and the acceleration data measurement could be placed near or away from the motor bearing. The experimental motor’s actual test conditions and bearing failure state were recorded in the data set.

In this experiment, we tested the diagnosis performance of the C/D-FUSA for the drive-end bearing fault data at 12,000 samples/s. The details of the tested data set are shown in Table 1. This experiment considered ten fault types, including one normal type (no-fault) and nine fault types. The nine fault types included three different locations (ball, inner raceway, outer raceway) and three fault diameters (0.007 mm, 0.014 mm, 0.021 mm). The details of the ten types are listed below:Normal type: no fault was found in these samples;Location = Ball, Diameter = 0.007: the fault occurred on the ball, the fault diameter was 0.007 in;Location = Ball, Diameter = 0.014: the fault occurred on the ball, the fault diameter was 0.014 in;Location = Ball, Diameter = 0.021: the fault occurred on the ball, the fault diameter was 0.021 in;Location = Inner Raceway, Diameter = 0.007: the fault occurred on the inner raceway, the fault diameter was 0.007 in;Location = Inner Raceway, Diameter = 0.014: the fault occurred on the inner raceway, the fault diameter was 0.014 in;Location = Inner Raceway, Diameter = 0.021: the fault occurred on the inner raceway, the fault diameter was 0.021 in;Location = Outer Raceway, Diameter = 0.007: the fault occurred on the outer raceway, the fault diameter was 0.007 in;Location = Outer Raceway, Diameter = 0.014: the fault occurred on the outer raceway, the fault diameter was 0.014 in;Location = Outer Raceway, Diameter = 0.021: the fault occurred on the outer raceway, the fault diameter was 0.021 in.

The numbers of samples with different types ranged from 487,093 to 2,182,450. To evaluate the performance of the proposed method under high- and low-frequency detection, we sampled 512 and 6000 consecutive and non-overlapping vibration signals as one sample and created new data sets: CWRU-512 and CWRU-6000, for training and diagnosis evaluation. The CWRU-512 contains 948–4261 samples for each category. The CWRU-6000 contains 80–360 samples for each category.

In the experiment, the number of categories $N_{C}$ was set as 10. The number of time steps T was set as 512 in CWRU-512 and T was 6000 in CWRU-6000. The subnet for the discontinuous features contained three convolutional modules and two LSTM layers were used in the subnet for the continuous features.

We conducted three comparisons in this work. In the first comparison, we compared the performance of the proposed C/D-FUSA with the end-to-end continuous/discontinuous feature fusion method without attention (C/D-FUS) and LSTM models. The C/D-FUS has the same structure as the C/D-FUSA, which did not use the context-dependent attention module for LSTM layers and SE attention modules for convolutional modules. The LSTM models contained two LSTM layers with 128 hidden cells. The features output in the last time step T were used for classification. Two fully connected layers followed the LSTM layers for classification. The first fully connected layer converted 128 features into 64 features. The last fully connected layer converted 64 features into 10 values. Like the C/D-FUSA, the LSTM model used the softmax function to convert the 10 values into 10 probabilities for each fault type. We used five-fold cross-validation for each model and each data set. We compared each method’s accuracy and loss curves and calculated the average accuracy, macro-precision, macro-recall and macro-f1-score for each model. In the second comparison, we compared the inference time for each model. Finally, in the last experiment, we compared the performance of C/D-FUSA with the state-of-the-art methods proposed in the other studies.

The accuracy, precision, recall and F1 scores were calculated in each data set. These measurements were calculated using Equations (21)–(24):(21)$$\mathrm{Accuracy}=\frac{TP+TN}{TP+TN+FN+FP}$$
(22)$$\mathrm{Precision}=\frac{TP}{TP+FP}\;$$
(23)$$\mathrm{Recall}=\frac{TP}{TP+FN}$$
(24)$$F1\;\mathrm{score}=\frac{2\times \mathrm{Precision}\times \mathrm{Recall}}{\mathrm{Precision}+\mathrm{Recall}}$$
where TP, TN, FN, and FP are the numbers of true positive, true negative, false negative and false positive samples.

The hyper-parameters used in all methods trained in the CWRU-512 or CWRU-6000 are the same, as shown in Table 2. The models for CWRU-512 and CWRU-6000 were trained on 200 epochs with an Adaptive Moment Estimation (Adam) optimizer [46] and cross-entropy loss function. The learning rates for CWRU-512 and CWRU-6000 were 0.002 and 0.001, respectively. Moreover, the max sequence length for these two data sets was 512 and 6000, respectively. The experiments were conducted using Python 3.6 (Python Software Foundation, Wilmington, DE, USA, http://www.python.org) and PyTorch 1.8 (Facebook AI Research, New York, NY, USA, https://pytorch.org/) on an Nvidia RTX 2060 GPU (Nvidia Corporation, Santa Clara, CA, USA).

### 3.2. Results

#### 3.2.1. Performance Comparison of Different Models

The accuracy and loss curves are shown in Figure 7 and Figure 8. It can be found that the LSTM model in CRWU-512 needed at least 30 times of training to improve the classification accuracy to about 90%, and this model in CRWU-6000 could not be well fitted even after 200 times of training. Compared with LSTM, the training performance of C/D-FUSA and C/D-FUS was significantly better, and the test classification accuracy of both data sets could stabilize at more than 95% after about 20 training sessions. However, the accuracy curve of C/D-FUS was less stable than that of C/D-FUSA, and the gap between the training and test accuracy of C/D-FUS was larger than that of C/D-FUSA, indicating that C/D-FUS was underfitting. In summary, through the analysis of the accuracy curves, it can be seen that the C/D-FUSA proposed by us is superior to the other two methods in terms of fitting speed, diagnosis performance and stability, no matter the data set.

We also calculated the accuracy, precision, recall and F1 scores for each model, shown in Table 3. According to this result, the diagnostic accuracy, precision, recall rate and F1 scores of C/D-FUSA were higher than the other methods in both CRWU-512 and CRWU-6000. This result indicates that the method proposed in this study performs better than the C/D-FUS method (the fusion of LSTM and CNN) and the LSTM method.

#### 3.2.2. Performance Comparison with Other Studies

We compared the performance of C/D-FUSA with other state-of-the-art methods in the CRUW data set, including Sohaib et al., (2017) [33], Le et al., (2019) [34], Lei et al., (2016) [47], Wang et al., (2022) [48], Yan et al., (2022) [49], Zhang et al., (2022) [50] and Zhao et al., (2021) [51]. The results are shown in Table 4. It can be found that the C/D-FUSA method proposed by us has better diagnostic performance than all other algorithms. The diagnostic model based on data preprocessing, LSTM, CNN and the attention mechanism had the best performance except for C/D-FUSA [34]. However, it required additional data processing, such as converting data to a frequency spectrum and envelope spectrum, and it also needed 5000 training epochs to achieve 99.74% accuracy. However, the C/D-FUSA is an end-to-end diagnostic algorithm, which does not require any complex data conversion or processing. This advantage would make it easier for use in different situations of bearing fault diagnosis. Moreover, it only needs 30–200 times of training to achieve 99.87% accuracy. The number of training samples differs between all of the methods because of the sampling frequency. This method achieves the highest accuracy when using a similar number of training samples as other methods.

#### 3.2.3. Inference Time Comparison with Other Studies

Although the performance of C/D-FUSA exceeds that of all methods compared in this work, the structure of C/D-FUSA is more complex than those of the C/D-FUS and LSTM models. Therefore, it is important that the inference speed of this model could be fast enough to be used in practical applications. We compared the inference time of the three models for a single sample, and the results are shown in Table 5. It can be found that although the inference time of C/D-FUSA for a single sample was longer than that of the other two models, the difference was not significant. Moreover, the single sample secondary sampling time of CRWU-512 is 0.043 s, and that of CRWU-6000 is 0.5 s. The inference time for one sample of C/D-FUSA is far shorter than the sampling time, indicating that the reasoning speed of the proposed algorithm can fully meet the requirements of practical use.

## 4. Discussion

In this work, we proposed an end-to-end continuous/discontinuous feature fusion method with CNN, LSTM and attention (C/D-FUSA) for rolling bearing fault diagnosis. We conducted experiments to test the diagnosis performance and the inference time of the proposed method.

In comparing the diagnosis performance, we compared C/D-FUSA with the other two simpler deep learning models and the state-of-the-art diagnosis methods in previous studies. According to the accuracy curves and loss curves of different models in the training process, it can be found that although the LSTM is found to be effective in some studies [52,53], these methods might require other optimization or extra data processing to be efficient. Therefore, the adaptability of the LSTM model is still insufficient, and it would be difficult to be well trained quickly in many situations. Therefore, using the LSTM model alone could not achieve an efficient end-to-end fault diagnosis. The performance curves of C/D-FUSA and C/D-FUS were excellent in both CWRU-512 and CWRU-6000. However, comparing the curves, it could be seen that C/D-FUSA was superior to C/D-FUS in terms of stability and training performance in the training process. The C/D-FUSA had extra attention modules compared with the C/D-FUS. This better performance indicates that both the context-dependent attention module proposed in this work and the SE attention module used in the convolutional modules effectively improve the diagnostic performance of the algorithm.

The excellent performance of C/D-FUSA was also proven in comparison with other advanced algorithms. Although some excellent work fused multiple signal processing, deep learning, transfer learning, and other advanced technologies, our work still had higher fault diagnosis performance than theirs [31,32,33,34,45,47,51]. In addition, these methods may need to transform the vibration signals to other spaces and use statistical methods to extract various features to achieve accurate fault diagnosis. On the one hand, these data-processing methods may consume more time, making it difficult to achieve real-time fault diagnoses of rolling bearings. On the other hand, they could be overly dependent on some designed features, which would be adjusted or modified for different rolling bearings. These disadvantages would make it difficult for these methods to be rapidly applied to a new work environment. Therefore, in an actual situation, the cost of these models may be higher. However, our proposed algorithm does not use complex data processing or feature engineering methods, and the deep learning model guides the whole diagnostic process. This end-to-end diagnostic approach can significantly reduce the difficulty and cost of model deployment and scenario migration in practice.

In addition to the performance of diagnostic accuracy, the method’s inference speed is also crucial. If the inference speed were too slow, it would be challenging to keep up with the signal sampling, making it unable to realize real-time fault diagnosis and, thus, miss some critical monitoring periods. The slow inference speed could bring missing faults and huge losses. We calculated the inference speed of our model at different diagnostic frequencies (inference per 512/6000 signals). The results showed that the inference speed of the proposed model was much faster than the signal-sampling frequency and was not significantly different from that of such simple model structures like LSTM and C/D-FUS. These results demonstrated that our proposed C/D-FUSA model could achieve the real-time, accurate, easily deployable and portable end-to-end rolling bearing fault diagnosis.

## 5. Conclusions

This paper proposed an end-to-end continuous/discontinuous feature fusion method for rolling bearing fault diagnosis (C/D-FUSA). This method comprises subnets for continuous features, discontinuous features and classification. The subnet for continuous features was composed of long short-term memory (LSTM) models with a proposed context-dependent attention module. This attention module weighted the final outputs of the LSTM according to the contextual outputs. This subnet was used for extracting the continuous features with contextual information. The subnet for discontinuous features was composed of several convolutional modules: a convolutional layer, a batch normalization layer, a ReLU activation function, a squeeze-and-excitation attention module and a max-pooling layer. The subnet was designed to extract features with discrete relations. Finally, the continuous and discontinuous features were input into the subnet for classification and transformed into probabilities of different fault types. We evaluated the proposed C/D-FUSA method in the data sets proposed by Case Western Reserve University with different sample rates. By comparing the diagnosis performance and inference time with the other, simpler deep learning methods and other state-of-the-art methods, the proposed method was proven to be accurate (99.85% of accuracy and 99.87% of F1 score) and in real time (0.034 s for one sample with 512 signals). In practical applications, our proposed model can be deployed in neural network chips or embedded systems, access multiple vibration sensor detection information, and directly output diagnostic information. The advantages of this method also make it easy to be deployed and migrated in practice.

## Figures and Tables

**Figure 1 sensors-22-06489-f001:**
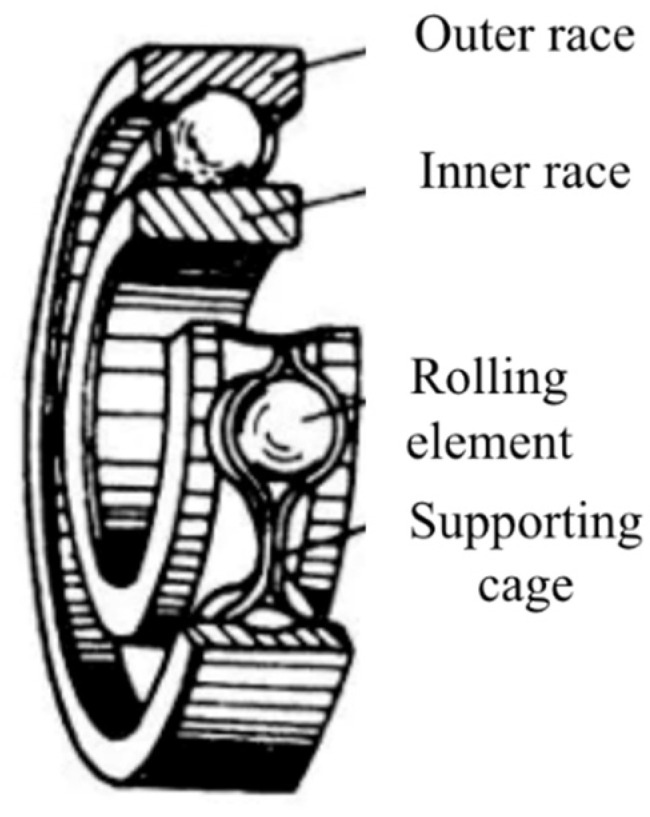
Rolling bearing structure, including outer race, inner race, rolling element and supporting cage.

**Figure 2 sensors-22-06489-f002:**
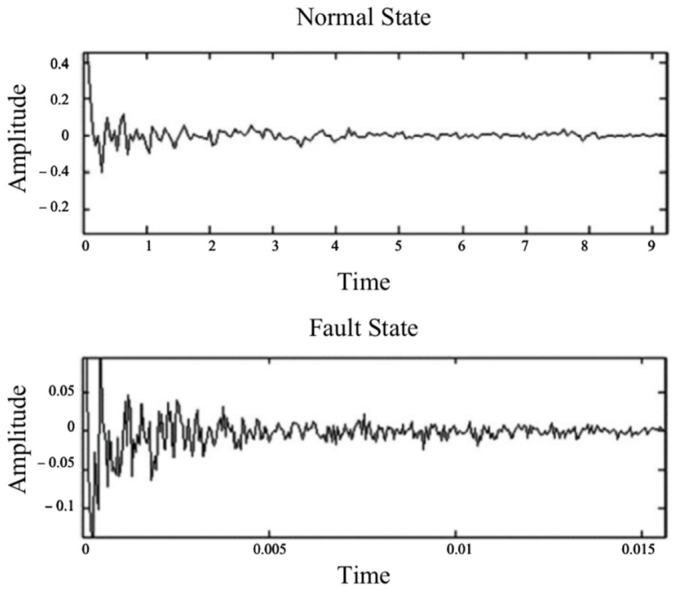
Signal cepstral analysis diagram for a normal state and a fault state.

**Figure 3 sensors-22-06489-f003:**
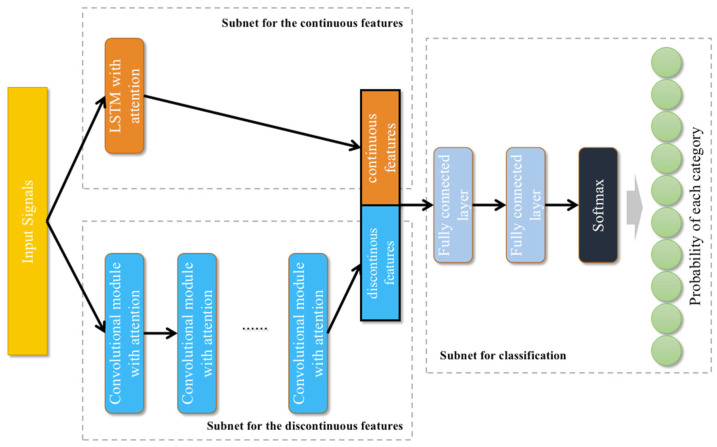
Framework for the proposed end-to-end continuous/discontinuous feature fusion method for rolling bearing fault diagnosis. This method contains three different subnetworks for the continuous features, the discontinuous features and classification. The subnetwork for discontinuous features used several convolutional modules with attention on extracting abstract discontinuous local features. The subnetwork for continuous features extracts the features used an LSTM layer with attention. The discontinuous and continuous features were concatenated and put into the subnetwork for classification, which contains several fully connected layers, to output the probability of each category.

**Figure 4 sensors-22-06489-f004:**
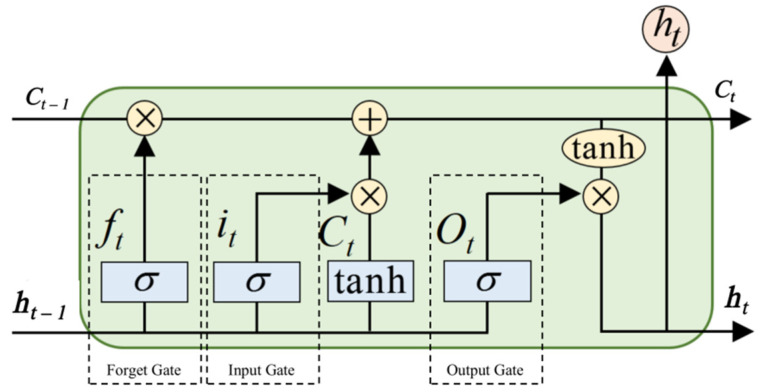
An LSTM cell.

**Figure 5 sensors-22-06489-f005:**
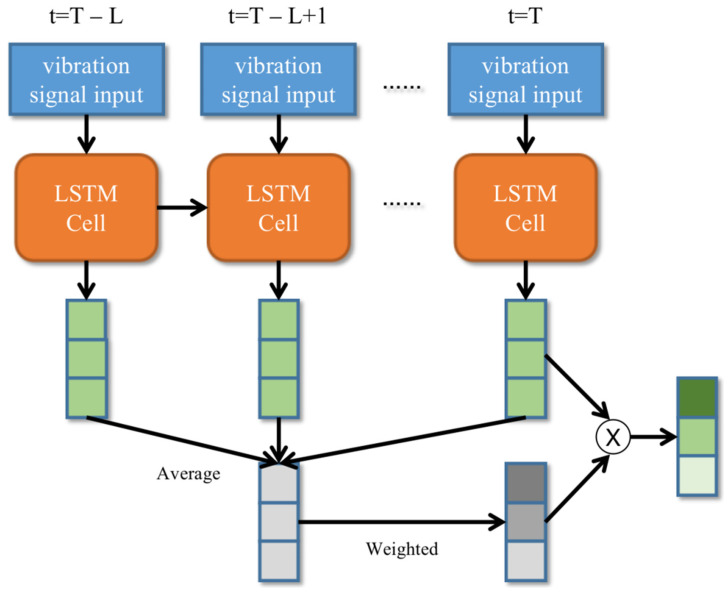
Context-dependent attention module for LSTM.

**Figure 6 sensors-22-06489-f006:**
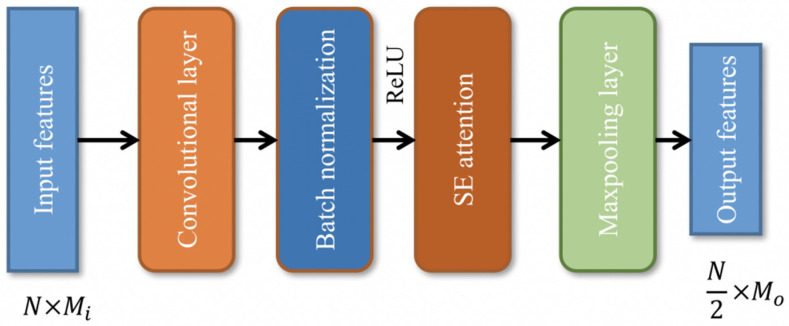
Structure of the convolutional modules.

**Figure 7 sensors-22-06489-f007:**
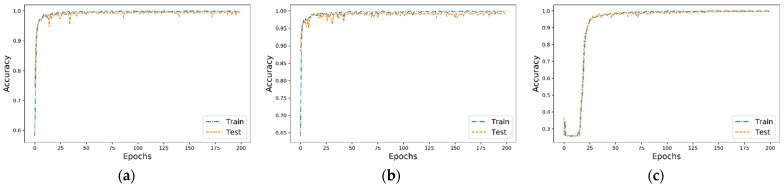
Accuracy curves for each model: (**a**–**c**) are the accuracy curves for C/D-FUSA, C/D-FUS and LSTM models for CWRU-512, respectively; (**d**–**f**) are the accuracy curves for C/D-FUSA, C/D-FUS and LSTM models for CWRU-6000, respectively.

**Figure 8 sensors-22-06489-f008:**
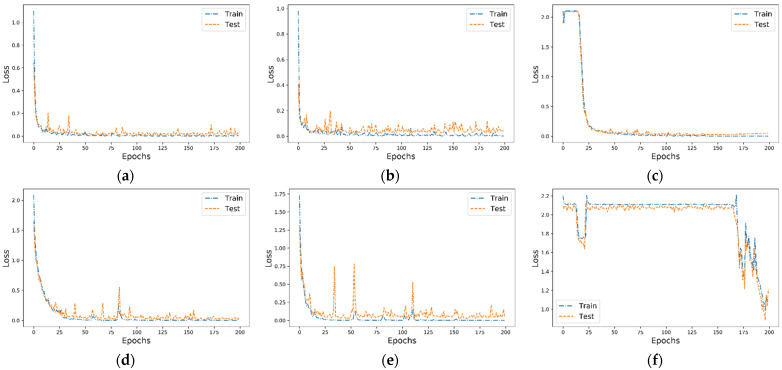
Loss curves for each model: (**a**–**c**) are the loss curves for C/D-FUSA, C/D-FUS and LSTM models for CWRU-512, respectively; (**d**–**f**) are the loss curves for C/D-FUSA, C/D-FUS and LSTM models for CWRU-6000, respectively.

**Table 1 sensors-22-06489-t001:** Details of the drive-end bearing fault data at 12,000 samples/s tested in our experiment.

Fault Location	Diameter	Number of Samples	Number of Samples in CWRU-512	Number of Samples in CWRU-6000
Normal	Normal	2,182,450	4261	360
Ball	0.007	487,093	950	80
	0.014	488,109	951	80
	0.021	487,964	951	80
Inner Raceway	0.007	488,309	952	80
	0.014	487,239	948	80
	0.021	487,529	950	80
Outer Raceway	0.007	1,465,051	2855	240
	0.014	487,819	950	80
	0.021	1,465,487	2856	240

**Table 2 sensors-22-06489-t002:** Hyper-parameters used in each data set.

Data Set	Hyper Parameter	Value
CWRU-512	Learning Rate	0.002
	Epoch	200
	Max length	512
	Optimizer	Adaptive Moment Estimation (Adam) [46]
	Loss Function	Cross-entropy loss
CWRU-6000	Learning Rate	0.001
	Epoch	200
	Max length	6000
	Optimizer	Adam
	Loss Function	Cross-entropy loss

**Table 3 sensors-22-06489-t003:** Diagnosis performance of each model in CRWU-512 and CRWU-6000.

Data Set	Model	Accuracy (%)	Precision (%)	Recall (%)	F1 Score (%)
CRWU-512	C/D-FUSA	99.85	99.84	99.90	99.87
	C/D-FUS	99.64	99.60	99.65	99.62
	LSTM	99.58	99.51	99.58	99.54
CRWU-6000	C/D-FUSA	99.69	99.65	99.72	99.68
	C/D-FUS	99.50	99.52	99.51	99.51
	LSTM	68.75	66.67	71.69	69.09

**Table 4 sensors-22-06489-t004:** Diagnosis performance comparison among the state-of-the-art methods and proposed method.

Method	Type	Number of Fault Classes	Number of Training Samples	Accuracy (%)
Sohaib et al., (2017) [33]	Hybrid features + sparse stacked autoencoder	10	710	99.10
Li et al., (2019) [34]	Preprocessing + attention + LSTM + CNN	10	-	99.74
Lei et al., (2016) [47]	Signal fraction + deep learning	10	20,000	99.66
Wang et al., (2022) [48]	SSAE and softmax classifier	10	4163	99.15
Yan et al., (2022) [49]	Markov transition field and residual network	10	6600	98.52
Zhang et al., (2022) [50]	Transfer learning	10	9518	99.80
Zhao et al., (2021) [51]	Adaptation network with adversarial learning	10	7000	99.24
C/D-FUSA	Attention + LSTM + CNN	10	13,299	99.85

**Table 5 sensors-22-06489-t005:** Inference time of each model in CRWU-512 and CRWU-6000.

Data Set	Model	Inference Time (s)
CRWU-512	C/D-FUSA	0.034
	C/D-FUS	0.031
	LSTM	0.028
CRWU-6000	C/D-FUSA	0.228
	C/D-FUS	0.220
	LSTM	0.208

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
