# Peer review of "End-to-End Continuous/Discontinuous Feature Fusion Method with Attention for Rolling Bearing Fault Diagnosis"

_sensors, 2022, doi:10.3390/s22176489_

Round 1
Reviewer 1 Report
Comments to authors
This paper proposes an end-to-end continuous/discontinuous feature fusion method for rolling bearing fault diagnosis (C/D-FUSA). This method comprises Long Short-Term Memory (LSTM), Convolutional Neural Networks (CNN) and 16 attention mechanism, which automatically extracts the continuous and discontinuous features from vibration signals for fault diagnosis. We also proposed a contextual-dependent attention module for the LSTM layers. We compared the method with the other simpler deep learning methods and state-of-the-art methods in rolling bearing fault data sets with different sample rates. But some comments should be taken into your consideration.
1- Editing English requires some effort to enhance your work.
2- You mentioned that ten fault types are carried out in the bearing system. Please state these fault types and the difference among them.
3- Table 4 compares your suggested model and other recently published works. The comparisons must be carried out with the methods that diagnose the same number of fault types so that the first two methods with 2 and 8 fault types must be removed from this table.
4- The comparison should be included the efficiency of each method with each fault type.
5- More detailed comparisons with recently published work are not considered in your results section. This comparison should include other factors than efficiency, such as time for training process, the number of samples considered for each method, etc.
Author Response
Thank you very much for your valuable comments,please see the attachment.

Reviewer 2 Report
Dear Authors,
I have some comments on your article:
1. It should be written how for FFT analyses there are important formulas contained in subsection: 2.1. Types of Rolling Bearing Faults. How do these equations translate into the characteristic features of the spectrum? Which are selected?
2. Subsection 2.2. Framework of the proposed C/D-FUSA – The algorithm presented in Figure 2. should be better described in the text: - Figure 2. Networks for the proposed end-to-end continuous/discontinuous feature fusion method for rolling bearing fault diagnosis. This method contains three different subnetworks for the continuous features, the discontinuous features and classification.
3. Subsection 3.1. Experiment setups – please provide more details on the measurements and data collected.
4. All indexes in symbols in text and equations should be checked carefully.
5. In the Conclusions section, please add information on how to apply the research contained in the article in practice - the development of a dedicated system with implemented deep learning algorithms.
6. Literature should be checked if there are no newer items. Especially from the last 18 months.
7. Please consider references to the spectral kurtosis in the introduction. Then I suggest adding to the list of literature:
- J. Antoni, "The spectral kurtosis: A useful tool for characterizing non-stationary signals", Mech. Syst. Signal Process., vol. 20, pp. 282-307, Feb. 2006.
- T. Barszcz R.B. Randall, “Application of spectral kurtosis for detection of a tooth crack in the planetary gear of a wind turbine,” Mechanical Systems and Signal Processing, 2009, Vol. 23, Issue 4, pp. 1352-1365, doi.org/10.1016/j.ymssp.2008.07.019.
- R. Zimroz and W. Bartelmus, "Application of adaptive filtering for weak impulsive signal recovery for bearings local damage detection in complex mining mechanical systems working under condition of varying load", Solid State Phenomena, vol. 180, pp. 250-257, Nov. 2011.
- P. Rzeszucinski, M. Orman, C. T. Pinto, A. Tkaczyk and M. Sulowicz, "Bearing Health Diagnosed with a Mobile Phone: Acoustic Signal Measurements Can be Used to Test for Structural Faults in Motors," in IEEE Industry Applications Magazine, vol. 24, no. 4, pp. 17-23, July-Aug. 2018, doi: 10.1109/MIAS.2017.2740463.
- M. Orman, P. Rzeszucinski, A. Tkaczyk, K. Krishnamoorthi, C. T. Pinto and M. Sulowicz, "Bearing fault detection with the use of acoustic signals recorded by a hand-held mobile phone," 2015 International Conference on Condition Assessment Techniques in Electrical Systems (CATCON), Bangalore, 2015, pp. 252-256, doi: 10.1109/CATCON.2015.7449545.
Author Response
Thank you very much for your valuable comments ,please see the attachment.

Round 2
Reviewer 1 Report
Thank you. All comments are answered; I accept the manuscript in this form.
Reviewer 2 Report
Dear Authors,
Thank you very much for introducing changes that have improved the quality of the article. I have no more comments.
Best regards